# Effect of laser photobiomodulation combined with hydroxyapatite nanoparticles on the osteogenic differentiation of mesenchymal stem cells using artificial intelligence: An *in vitro* study

Eloiza Leonardo de Melo[1], Jéssica Meirinhos Miranda[1]*, Vanessa Bastos de Souza Rolim Lima[1], Wyndly Daniel Cardoso Gaião[2], Braulio de Vilhena Amorim Tostes[3], Claudio Gabriel Rodrigues[2], Márcia Bezerra da Silva[2], Severino Alves Júnior[3], Edson Luiz Pontes Perger[4], Mávio Eduardo Azevedo Bispo[5], Marleny Elizabeth Márquez de Martínez Gerbi[1]

1 Department of Biophotonics in Health Sciences, University of Pernambuco, Recife, Pernambuco, Brazil,
2 Department of Biophysics and Radiobiology, Federal University of Pernambuco, Recife, Pernambuco, Brazil, 3 Department of Basic Chemistry, Federal University of Pernambuco, Recife, Pernambuco, Brazil, 4 Department of Bioprocess and Biotechnology, Paulista State University, Botucatu, São Paulo, Brazil, 5 Department of Computer Science, University of Pernambuco, Recife, Pernambuco, Brazil

* jessica.meirinhos@upe.br

## Abstract

### Aim

To evaluate *in vitro* the effect of laser photobiomodulation (PBM) combined or not with 30-nm hydroxyapatite nanoparticles (HANp), on the osteogenic differentiation of human umbilical cord mesenchymal stem cells (hUC-MSCs) by morphometric analysis using artificial intelligence programs (TensorFlow and ArcGIS).

### Methods

UC-MSCs were isolated and cultured until 80% confluence was reached. The cells were then plated according to the following experimental groups: G1 –control (DMEM), G2 –BMP-2, G3 –BMP-7, G4 –PBM (660 nm, 10 mW, 2.5 J/cm², spot size of 0.08 cm²), G5 –HANp, G6 –HANp + PBM, G7 –BMP-2 + PBM, and G8 –BMP-7 + PBM. The MTT assay was used to analyze cell viability at 24, 48 and 72 h. Osteogenic differentiation was assessed by Alizarin Red staining after 7, 14 and 21 days. For morphometric analysis, areas of osteogenic differentiation (pixel²) were delimited by machine learning using the TensorFlow and ArcGIS 10.8 programs.

### Results

The results of the MTT assay showed high rates of cell viability and proliferation in all groups when compared to control. Morphometric analysis revealed a greater area of osteogenic

**Data Availability Statement:** All relevant data are within the manuscript. However, We are available to provide any additional data.

**Funding:** ELM - This research was supported by a doctoral scholarship provided by the Coordination for the Improvement of Higher Education Personnel (CAPES/ process number: 88882.435699/2019-01) https://www.gov.br/capes/pt-br JMM - This research was supported by a doctoral scholarship provided by the Coordination for the Improvement of Higher Education Personnel (CAPES/ process number: 88882.435693/2019-01) https://www.gov.br/capes/pt-br MEMMG - by the Pernambuco Science and Technology Foundation (FACEPE-APQ-0886-4.02/10) https://www.facepe.br/ and by the APQ research support call (573-PFA/UPE) of the University of Pernambuco (UPE) / https://www.upe.br/pfa2.html The sponsors or funders had no role in the study design, data collection and analysis, decision to publish, or preparation of the manuscript.

**Competing interests:** The authors have declared that no competing interests exist.

differentiation in G5 (HANp = 142709,33±36573,39) and G6 (HANp + PBM = 125452,00 ±24226,95) at all time points evaluated.

## Conclusion

It is suggested that HANp, whether combined with PBM or not, may be a promising alternative to enhance the cellular viability and osteogenic differentiation of hUC-MSCs.

## Introduction

Considering the increase in life expectancy and the growing need for bone repair, the development of new techniques for the reconstruction of bone defects is currently a field of interest in tissue engineering and regenerative medicine. These areas use biomaterials to stimulate/induce *in vitro* the differentiation of undifferentiated stem cells into specialized cells or tissues [1]. Within this context, different strategies are employed to stimulate cellular activity, including photobiomodulation (PBM) and the inclusion of nanoparticles in cell culture media [2].

When administered adequately, PBM exerts photochemical and/or photobiological effects that lead to the provision of energy to the cell, stimulating cellular functions such as cell migration, mitochondrial ATP production, tissue repair, cell proliferation and differentiation, and collagen synthesis [3]. In tissue regeneration, the applications of PBM have provided good results, especially in terms of osteogenic and angiogenic differentiation [4,5].

Nanoparticles are characterized by their nanometric size, easy synthesis and surface functionalization, as well as their bioactivity, biodegradability, and biocompatibility. In medicine, nanoparticles are used for drug delivery and dental tissue regeneration. Hydroxyapatite nanoparticles (HANp) are particularly interesting because hydroxyapatite is a component of mineralized tissues and these particles may therefore be able to guide stem cells towards osteogenic differentiation [6–8].

For the *in vitro* analysis of the activity of these inductions, methods are implemented to quantify cellular activity. Artificial intelligence through machine learning supports analyses in the health area by increasing speed and accuracy and reducing human errors [9].

In view of the lack of studies combining PBM and HANp for the induction of osteogenic differentiation of human umbilical cord mesenchymal stem cells (UC-MSCs), this study aimed to evaluate the influence of PBM (660 nm, 10 mW, 2.5 J/cm$^2$, spot size of 0.08 cm$^2$), combined or not with 30-nm HANp, on cell proliferation (MTT) and on osteogenic differentiation evaluated by morphometric analysis using artificial intelligence (programas TensorFlow e ArcGIS 10.8).

## Materials and methods

### Ethical considerations

This study followed all local statutory guidelines and was approved by the Ethics Committees of the University of Pernambuco (CAAE 47073121.9.0000.5207) and of the co-participating institution (Federal University of Pernambuco; CAAE 47073121.9.3002.5208). The study was conducted in accordance with the Declaration of Helsinki. The type of consent for umbilical cord collection was in writing and all umbilical cord donors signed the informed consent form. The pregnant women who participated in the research were over 18 years of age (with 36 to 40 weeks of gestational age), with no systemic alterations, infectious diseases, or exposure

to chemotherapy/radiotherapy in the last five years. This study began on February 10, 2022 and was completed on February 2, 2024.

## Isolation and culture of hUC-MSCs

Human umbilical cord was collected immediately after cesarean delivery and transferred to a glass jar containing sterile phosphate-buffered saline (PBS), 3 mM ethylenediamine tetraacetic acid (EDTA), 200 IU/mL penicillin, 200 g/mL streptomycin, and 5 μg/mL amphotericin B. The umbilical cord stroma was extracted, cut into pieces of approximately 0.5 cm, and placed in cell culture bottles containing 10 mL Dulbecco's Modified Eagle's Medium (DMEM) (Gibco, ThermoFisher, São Paulo, Brazil) supplemented with 15% fetal bovine serum (FBS; Gibco, ThermoFisher, São Paulo, Brazil), 20% Ham F-12 (Gibco, ThermoFisher, São Paulo, Brazil), and antibiotics.

The bottles were kept in an incubator at 37°C in a 5% $CO_2$ atmosphere until the cells reached 80% confluence (third passage). The cells were phenotyped by flow cytometry (FACS-calibur, BD Bioscience, San Jose, CA, USA) to confirm their mesenchymal nature. For subsequent counting and plating, the cells were incubated with fluorescent CD90, CD44 and CD29 monoclonal antibodies (EXBIO Antibodies, Prague, Czech Republic) conjugated with fluorescein isothiocyanate, and with CD45, CD34 and CD31 antibodies conjugated with phycoerythrin (diluted: 1:2000) for 60 minutes at 4°C.

Volumes of cell suspensions containing $5 \times 10^4$ and $1 \times 10^4$ cells/mL were placed in the wells of 24-well and 96-well plates, respectively. The culture plates were kept in an incubator at 37°C in a 5% $CO_2$ atmosphere for 24 h before the experiment [5].

## Preparation of media for inducing osteogenic differentiation

Twenty-four hours after cell plating, the inducers of osteogenic differentiation were added: HANp, bone morphogenic proteins (BMP) 2 and 7, and PBM. The control group consisted of DMEM medium supplemented with 15% fetal bovine serum (FBS; Gibco ThermoFisher,São Paulo, Brazil), 20% Ham F-12 (Gibco, ThermoFisher, São Paulo, Brazil), and antibiotics. The HANp with a mean size of 30 nm were added to the osteogenic medium at a concentration of 25 μg/mL. The concentration was chosen based on the study by Remya et al. [10].

BMP-2 and BMP-7 (Gibco) at a final concentration of 10 ng/mL were added to DMEM (Gibco, ThermoFisher, São Paulo, Brazil) supplemented with 10% FBS (Gibco, ThermoFisher, São Paulo, Brazil), 50 mg/mL PSG (Invitrogen, ThermoFisher, Waltham, Massachusetts, EUA), 50 μg ascorbic acid (Sigma Aldrich, São Paulo, Brazil), 1 mM β-glycerophosphate (Sigma Aldrich, São Paulo, Brazil), and 10 nM dexamethasone (Sigma Aldrich, São Paulo, Brazil).

## Irradiation of hUC-MSCs

Alternate wells of the plates were irradiated. Before PBM, the medium of the wells was temporarily replaced with PBS, which is a transparent substance that allows the passage of light. The plates were then covered with a black card containing only one hole according to the location of the wells. Only the area to be irradiated was exposed in order to avoid the irradiation of adjacent wells and energy overload.

The PBM experiments were standardized using a claw-shaped support to maintain the laser's tip in a single position and a previously fabricated stand in order to fix the distance between the bottom of the well and the tip of the laser at 6 cm. The cells of the PBM groups were irradiated with a low-level InGaAlP laser (MMOptics® Equipamentos Ltda, São Carlos, SP, Brazil) using the following parameters: 660 nm, 10 mW, 2.5 J/cm², and spot size of 0.08

**Table 1. Irradiation parameters.**

| Irradiation Parameters | |
|---|---|
| Emission Mode (CW) | Continuous |
| Wavelength | 660nm |
| Active Medium InGaAIP | InGaAIP |
| Laser Optical Power (Output) | 10mW |
| Laser Optical Power (Input) | 10mW |
| Power Density (PD) | 10mW |
| Energy Density (ED) | 2,5J/cm$^2$ |
| Irradiance | 10 mW |
| Spot | 0,08cm$^2$ |
| Time per Well | 10s |

cm$^2$ (Table 1). This laser was chosen because it is well-established in the literature that it accelerates the regenerative process of tissues due to its biomodulatory effect, as evidenced by a previous study showing positive results after using this laser on hUC-MSCs [5].

## Experimental groups

The groups were divided into 8, as follows: Group 1 (G1) corresponded to the Control (the wells with hUC-MSCs received only the control medium (DMEM); Group 2 (G2) corresponded to the group where the wells with hUC-MSCs received BMP-2 as the induction medium; Group 3 (G3) corresponded to the group where the wells with hUC-MSCs received BMP-7 as the induction medium; Group 4 (G4) corresponded to the group where the wells with hUC-MSCs received PBM as the induction medium; Group 5 (G5) corresponded to the group where the wells with hUC-MSCs received HANp as the induction medium; Group 6 (G6) corresponded to the group where the wells with hUC-MSCs received a combination of HANp and PBM as the induction medium; Group 7 (G7) corresponded to the group where the wells with hUC-MSCs received a combination of BMP-2 and PBM as the induction medium; and Group 8 (G8) corresponded to the group where the wells with hUC-MSCs received a combination of BMP-7 and PBM as the induction medium. The experimental groups were organized according to Table 2.

## Experimental design

Fig 1 illustrates the steps carried out in this study. After confirming the nature of the hUC-MSCs, they were plated and divided into eight groups according to the induction media.

**Table 2. Experimental groups.**

| Group | Inducing medium |
|---|---|
| G1 | DMEM (control) |
| G2 | BMP-2 |
| G3 | BMP-7 |
| G4 | PBM |
| G5 | HANp |
| G6 | HANp + PBM |
| G7 | BMP-2 + PBM |
| G8 | BMP-7 + PBM |

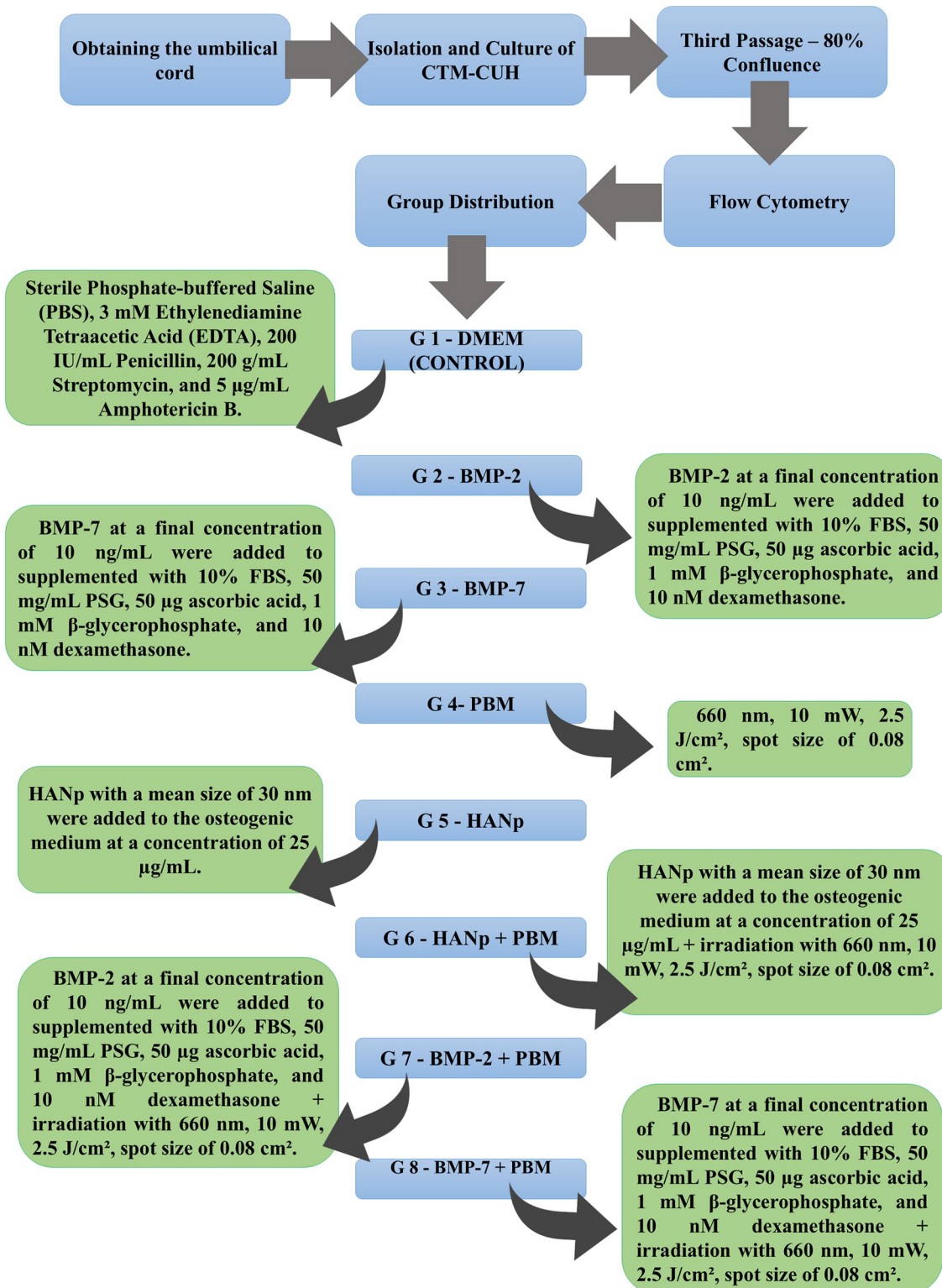

**Fig 1. Experimental design.**

Subsequently, assays were conducted to evaluate viability (MTT assay) and cellular differentiation in all groups.

## MTT assay

Cell viability was analyzed in the groups at intervals of 24, 48 and 72 h for construction of the cell proliferation curve. For the assay, 0.5 mg of 3-(4,5-dimethyl-2-thiazolyl)-2,5-diphenyl tetrazolium bromide (MTT)/mL medium was added to the wells and the plates were incubated in an oven for 4 h at 37°C, protected from light. After incubation, MTT was removed from the wells and dimethyl sulfoxide (DMSO; Sigma) was added to solubilize the formazan crystals formed. Absorbance was read in a spectrophotometer (FLX 800 Fluorescence Microplate Reader, software version 2.06.10, BIOTEK, Winooski, VT, USA) at a wavelength of 495 nm. The experiments were carried out in triplicate.

## Cell differentiation assay

Morphological analysis was performed 7, 14 and 21 days after the induction of osteogenic differentiation. After the period of cell culture in the inducing media, each well was washed three times with PBS until complete removal of the medium. The cells were fixed in 300 μL of 10% paraformaldehyde for 20 minutes at room temperature, followed by staining with 300 μL Alizarin Red for 15 minutes, a specific dye that detects calcium deposits. The wells were washed again thoroughly with deionized water to remove excess dye. Images of the bottom of the wells of each experimental group were captured with a camera coupled to an inverted optical microscope (Leica DM1000, Leica, Wetzlar, Germany) at magnifications of 40X and 100X. The experiments were carried out in triplicate.

## Machine learning using the TensorFlow and ArcGIS programs

The same images as employed in the cell differentiation assay were used in this step. The TensorFlow and ArcGIS programs were used for the identification, measurement, and quantification of the mineralized tissue recognized by AI (machine learning), as preliminary tests were conducted by feeding data into the software for the identification of mineralized tissue already confirmed in the cellular differentiation assay. For morphometry of the differentiation areas using the TensorFlow software, the semantic segmentation technique was applied as described by Shelhamer et al. [11].

First, the images were resized to 640×480 pixels and transformed to.png format to meet the requirements of the software used. For creation of the dataset, the areas of interest were marked (painted) in magenta (RGB = 255, 0, 255) using the Inkscape program and a script in Python to check the pixels with RGB of the required shape. The dataset was then loaded into TensorFlow, which replaced the areas of interest (in magenta) with white color. Finally, the pixels were obtained and stored in an Excel 2010 spreadsheet.

Thus, TensorFlow, in conjunction with convolutional neural networks, creates models that learn to identify hierarchies of features in images, ranging from simple edges to complex biological structures. The images are transformed into feature maps that highlight relevant regions. The neural network then reduces the dimensionality of these maps by selecting the most prominent patterns and generates a pixel-by-pixel segmentation mask. The model's accuracy is evaluated using the IoU (Intersection over Union) metric, which compares the model's segmentation with the reference segmentation, yielding a value of 0.94 in this case, indicating high precision of the results.

For analysis with the ArcGIS program, the ArcMap version 10.8 was used. The images were classified using the Iso Cluster Unsupervised Classification tool, with 50% transparency. Red

was used for areas of interest and black for areas of no interest. The number of pixels was then obtained and entered into a previously created Excel 2010 spreadsheet. The images were extracted and saved in specific folders for each group.

The ArcGIS program uses RMSE (Root Mean Square Error) to measure the accuracy of predicted values. RMSE should be compared with observed values to assess the model's quality. The Spread Index (SI) is calculated by dividing the RMSE by the observed mean value and multiplying by 100%. An SI of less than 10% indicates a good model, and less than 5% indicates a very good model. According to the Ashrae standard, a forecasting model should have an $R^2$ (coefficient of determination) greater than 0.75 and an SI below 30% for annual data and 10% for hourly or monthly data. In terms of RMSE, smaller values are better. An $R^2$ of 0.999 and RMSE of 0.1 or less are considered very satisfactory. In this study, the $R^2$ was 0.95 and the RMSE was 0.05 or less, which are considered very satisfactory.

## Statistical analysis

The data of the MTT assay and of the morphometric analysis of cell differentiation using TensorFlow and ArcGIS were evaluated by calculating the mean of the values obtained for each well of each group. The Kruskal-Wallis test was used to evaluate the experimental groups at each time. The Friedman test was applied for comparison between time in each group. Statistical significance was set at 5% ($p < 0.05$) and the analyses were performed using the Rproject program.

## Results

### MTT assay

Fig 2 shows the cell proliferation curve, expressed as optical density, at 24, 48 and 72 h. As can be seen, the groups exhibited high rates of cell proliferation at all time points evaluated when compared to G1 (control), apresentando diferença estatística significativa. It was observed that in the first 24 hours, cellular proliferation was higher in the BMP-2 and HANp groups, although there was no statistically significant difference between them. When evaluated at 48 hours, the group that exhibited the highest cellular proliferation was the HANp+PBM group, followed by the HANp and BMP-2 groups, again without a statistically significant difference among them. After 72 hours, the highest level of cellular proliferation was attributed to the HANp group, while the lowest levels were associated with the control, BMP-2, and BMP-7 groups.

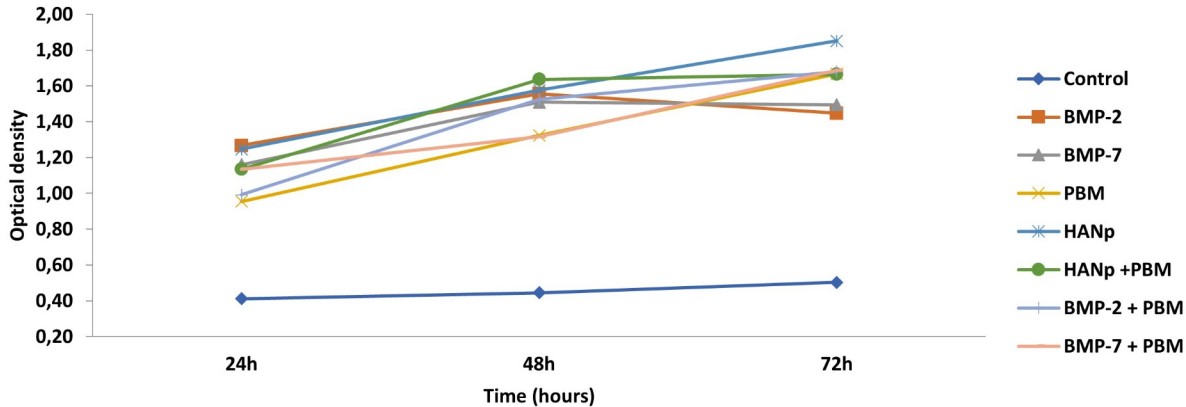

**Fig 2. Line graph highlighting the cellular proliferation curve (optical density) in the control group and experimental groups (G2: BMP-2; G3: BMP-7; G4: PBM; G5: HANp; G6: HANp + PBM; G7: BMP-2 + PBM; G8: BMP-7 + PBM) at 24, 48, and 72 hours.**

## Cell differentiation assay

Morphological analysis (Fig 3) showed intense osteogenic activity 7 days after induction, as demonstrated by the formation of calcium nodules (red areas). Higher intensity was observed in G6 (HANp + PBM) and G5 (HANp), followed by G8 (BMP-7 + PBM), G7 (BMP-2 + PBM), and G2 (BMP-2). No osteogenic differentiation was detected in G1, G3 or G4 after this period.

The images also revealed agglomerates in the central regions (black arrows), associated with more pronounced differentiation (intense red color), in G6 (more evident) and G5. In addition, there was a triangular-shaped structure (yellow arrow) in image P of Fig 3, a finding that might be related to the presence of HANp and their internalization inside the cells.

Intense osteogenic activity was found 14 days after induction, as demonstrated by red-stained regions, which were more evident in G4 (PBM), followed by G2 (BMP-2), G5 (HANp), G6 (HANp + PBM), G3 (BMP-7), G7 (BMP-2 + PBM), and G8 (BMP-7 + PBM). No osteogenic activity was detected in the control group (Fig 3).

Twenty-one days after induction, osteogenic activity remained only in G6 (HANp + PBM) and G5 (HANp). A considerable decline in osteogenic activity was observed in G3 (BMP-7), G8 (BMP-7 + PBM), G7 (BMP-2 + PBM), G4 (PBM), and G2 (BMP-2) when compared to G1 (control), which did not exhibit cell differentiation. Central regions with intense red staining (yellow arrows) were detected in G5 and G6 but not in the other groups (Fig 3).

## Morphometric analysis using TensorFlow and ArcGIS

Fig 4 illustrates an example of each image after application of the TensorFlow and ArcGIS programs. The image shows the incomplete delimitation (red arrows) of the areas of osteogenic differentiation by the TensorFlow program (b), while the ArcGIS program completely delimited the entire area of osteogenic differentiation (d).

Fig 5 shows the statistical analysis of the area of osteogenic differentiation, in pixel$^2$, according to group and time after induction using the TensorFlow program. It was observed that after 7 days of osteogenic differentiation, the group with the highest values of cellular differentiation was the HANp+PBM group, while the lowest values were associated with the control, BMP-7, and PBM groups. When evaluated at 14 days, the highest osteogenic differentiation values were presented by the PBM, HANp, and BMP-2 groups, with no statistically significant difference among them. The lowest value was associated with the control group. After 21 days of osteogenic differentiation, the highest values were attributed to the HANp and HANp +PBM groups, while the lowest values were related to the control, BMP-2, and PBM groups, with no statistically significant difference among them. The mean and standard deviation values are available in Table 3.

Fig 6 shows the statistical analysis of the area of osteogenic differentiation, in pixel$^2$, according to group and time after induction using the ArcGIS program. It was observed that at 7 days of osteogenic differentiation, the groups with the largest differentiated areas were the HANp+PBM and HANp groups, which were statistically significant. The lowest values were associated with the control, BMP-7, and PBM groups, with no statistically significant difference among them. When evaluated at 14 days of osteogenic differentiation, the groups with the highest cellular differentiation values were the PBM and BMP-2 groups, with no statistically significant difference between them. The lowest cellular differentiation area was associated with the control group. After 21 days, the largest cellular differentiation area was attributed to the HANp and HANp+PBM groups, with no statistically significant difference between them. The lowest values were related to the control, BMP-2, and PBM groups, with no statistically significant difference among them. The mean and standard deviation values are available in Table 4.

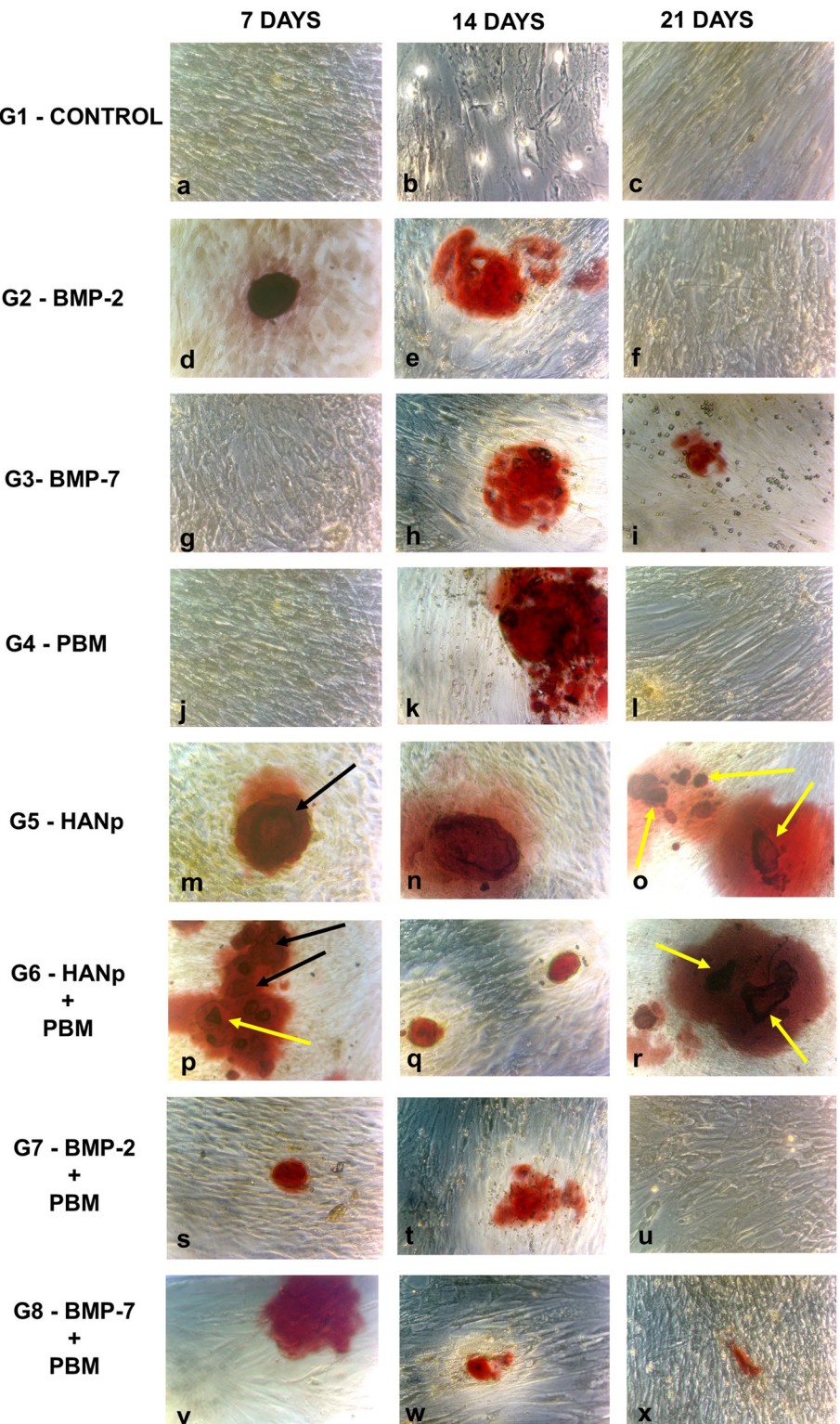

**Fig 3. Morphological analysis.** Inverted phase contrast images obtained for all groups after 7, 14 and 21 days (40X magnification).

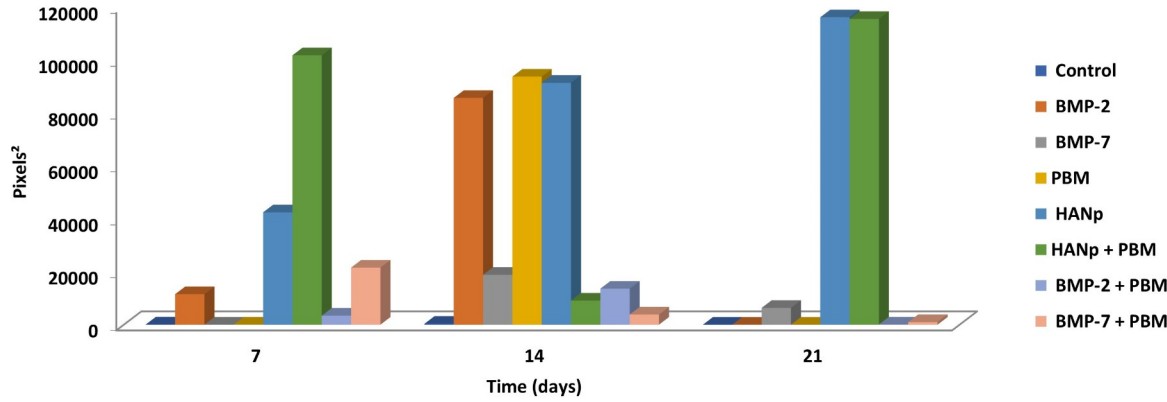

**Fig 4.** (a) Morphological analysis. Inverted Optical Microscope image (40X magnification). (b Image showing osteogenic differentiation (areas of margenta color) (40X magnification). (cImage showing osteogenic differentiation. Delimitation of areas of interest (white) obtained by machine learning (40X magnification). (d) Images showing delimitation of the areas of interest (red) after osteogenic induction obtained with the ArcGIS program.

## Discussion

Several approaches are used to promote increased cellular activity and accelerate bone regeneration, including PBM and nanoparticles added to cell culture media. Thus, the aim of this study was to evaluate in vitro the effect of laser photobiomodulation (PBM) combined or not with 30-nm hydroxyapatite nanoparticles (HANp), on the osteogenic differentiation of human umbilical cord mesenchymal stem cells (hUC-MSCs) by morphometry using artificial intelligence programs (TensorFlow and ArcGIS). The results showed that group 5, which used only 30 nm of HANp, and group 6, which combined HANp with PBM, significantly increased cell viability, cell proliferation, and osteogenic differentiation.

The present study makes two main contributions to tissue regeneration. First, according to Zhang et al. [1], with the increase in life expectancy, it is important to develop new protocols and biomaterials for the repair of bone defects since the available biomaterials may have limitations and exert undesirable effects on the regeneration of mineralized tissues [12]. Second, the study highlights the contribution of artificial intelligence and machine learning to scientific research, with results comparable to those obtained with methodologies that are well established in the literature.

Photobiomodulation stimulates cellular activity, increasing proliferation and osteogenic differentiation through its biomodulatory effect [5,13,14]. The effects of this technique depend on different factors such as wavelength, energy density, irradiation frequency, and focal distance of the laser [15]. The protocol used here was based on a previous study that reported positive results [5].

**Fig 5. Bar graph showing the pixel² values obtained with the TensorFlow program for all groups according to time after osteogenic induction.**

**Table 3. Analysis through TensorFlow regarding the groups and time points.**

| Groups | 7 Days | 14 Days | 21 Days | p-value§ |
|---|---|---|---|---|
| Control | 101,33±155,89Fb | 333,00±265,03Ea | 16,67±28,87Ec | <0,0001 |
| BMP-2 | 11548,67±11027,88Db | 85728,33±32845,49Aa | 26,00±2,65Ec | <0,0001 |
| BMP-7 | 44,00±34,69Fc | 18849,33±25632,59Ba | 6354,33±7156,35Bb | <0,0001 |
| PBM | 59,67±85,80Fb | 93776,33±20232,18Aa | 7,67±11,59Ec | <0,0001 |
| HANp | 42487,00±16551,26Bc | 91463,67±36451,20Ab | 116209,67±21553,56Aa | <0,0001 |
| HANp + PBM | 101837,33±32945,03Aa | 9127,00±3699,39Cb | 115600,33±14063,21Aa | <0,0001 |
| BMP-2 + PBM | 3425,67±4491,99Eb | 13629,00±7566,37Ba | 86,67±130,61Dc | <0,0001 |
| BMP-7 + PBM | 21589,33±27290,85Ca | 3832,33±3369,38Db | 998,00±1662,81Cc | <0,0001 |
| p-value* | <0,0001 | <0,0001 | <0,0001 | |

Capital letters compare in column, lower case letters compare in row.

* Kruskal-Wallis test.

§ Friedman test.

The current study used a wavelength of 660 nm, which provided positive results in terms of cell proliferation. This wavelength show good absorption by cytochrome c oxidase as the main photon acceptor in mitochondria [16]. After photoabsorption, the increase in electron transport chain activity and the initiation of a cellular signaling cascade can stimulate cell proliferation and differentiation [4].

Low energy and power density values were used in this study, in agreement with the parameters reported by Ginane et al. [17] who used 1 J/cm$^2$ and 30 mW, respectively. These results are understandable since an excessive increase in these parameters can damage the photoreceptors, with a consequent reduction in the biomodulatory effect of the laser [18]. Furthermore, the heterogeneity in parameters impairs the comparison between studies.

Bone morphogenetic proteins are the gold standard growth factors for tissue regeneration [19], confirming the findings of Zhang et al. [20]. However, the high cost of these proteins limits their application and justifies the search for other experimental therapies such as HANp and/or PBM. In this context, the present study observed that HANp, whether associated with

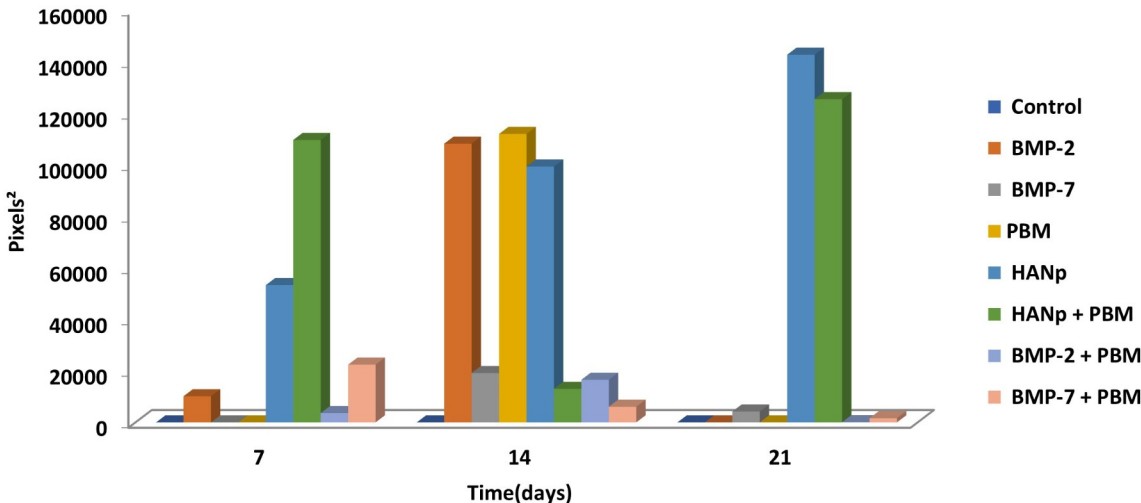

**Fig 6. Bar graph showing the pixel$^2$ values obtained with the ArcGIS program for all groups according to time after osteogenic induction.**

**Table 4. Analysis through ArcGIS regarding the groups and time points.**

| Groups | 7 Days | 14 Days | 21 Days | p-value§ |
|---|---|---|---|---|
| Control | 0,00±0,00Fa | 0,00±0,00Ea | 0,00±0,00Ea | 1,000 |
| BMP-2 | 10148,67±8821,54Da | 108177,67±53140,18Aa | 0,00±0,00Eb | <0,0001 |
| BMP-7 | 0,00±0,00Fc | 19174,67±24791,43Ca | 4232,67±3799,82Bb | <0,0001 |
| PBM | 0,00±0,00Fb | 111957,67±20281,16Aa | 0,00±0,00Eb | <0,0001 |
| HANp | 53290,67±20480,86Bc | 99303,67±50502,97Bb | 142709,33±36573,39Aa | <0,0001 |
| HANp + PBM | 109627,00±32444,31Aa | 13028,33±2169,29Cb | 125452,00±24226,95Aa | <0,0001 |
| BMP-2 + PBM | 3613,67±4233,12Eb | 16554,33±2169,29Ca | 78,00±124,85Dc | <0,0001 |
| BMP-7 + PBM | 22423,33±26485,83Ca | 6087,00±3830,14Db | 1639,67±2839,99Cc | <0,0001 |
| p-value* | <0,0001 | <0,0001 | <0,0001 | |

Capital letters compare in column, lower case letters compare in row.

* Kruskal-Wallis test.

§ Friedman test.

PBM or not, resulted in increased proliferation and osteogenic differentiation, yielding results as promising as those observed in other studies that utilized only BMPs as induction factors [19,20].

Nanomaterials such as HANp are widely studied because of their osteoinductive and osteo-conductive properties [21], which were also demonstrated by morphological analysis in the present study. Osteogenic differentiation was maintained throughout the periods studied. Furthermore, HANp can be synthesized rapidly and at low cost when compared to BMPs. This technology opens up possibilities for future research. We therefore recommend conducting further studies that focus on osteogenic tissue regeneration using HANp or that employ these nanoparticles as carriers of growth factors in the culture medium or even as markers of cellular activity.

The MTT assay showed an increase in cellular viability after 48 and 72 hours in groups G6 (HANp + PBM) and G5 (HANp), enhancing the potential for cellular proliferation. These results are consistent with the findings reported by Miranda et al. [5], who used the same cell viability assay, although cellular differentiation was stimulated by PBM and osteogenic medium without the use of HANp. Findings reported by Liu et al. [22] indicated that HANp could promote the growth of mesenchymal stem cells derived from bone marrow when the particle concentrations were below 20 μg/10^4 cells. These results align with the findings by Remya et al. [10] and Shi et al. [23], which demonstrated that these nanoparticles are not toxic to stem cells. In conclusion, the two experimental methods used to stimulate cellular proliferation (HANp and PBM), applied either alone or in combination, maintain cellular viability while enhancing proliferation potential, thereby exhibiting biomodulatory effects without harming the cells.

Morphological analysis after 7 days showed the internalization of HANp into the cells. Similar findings have been reported by Shi et al. [23]; however, the authors monitored this internalization based on transmission electron photomicrographs obtained after 1, 2 and 4 h. The geometric image seen in Fig 2 (P, yellow arrow) might be related to larger hydroxyapatite particles. After calcination, the material is ground to produce nanoparticles; however, particles of varying sizes still remain. Thus, alternatives for better refinement of the material are needed in order to obtain uniform sizes.

The HANp group showed a progressive increase in osteogenic differentiation from days 7 to 21, corroborating the study by Shi et al. [23], which indicated their stimulatory capacity for

osteogenesis, suggesting that HANps promote the differentiation of osteoblasts and the production of extracellular matrix. The decrease in mineralization from day 14 to day 21 in the groups that did not contain HANp may be attributed to the gradual reduction in the self-renewal capacity and the subsequent specialization of the cells into functional cells, in this case, towards osteoblasts. Additionally, the HANp groups demonstrated a positive effect on the adhesion and differentiation of stem cells and exhibited excellent biological properties compared to the other analyzed groups [24].

The application of automated computational methods to the measurement of cell differentiation areas in the field of tissue regeneration is expanding. Shouval et al. [25], Zhang et al. [26], Zaman et al. [27], Coronnello et al. [28], and Juhola et al. [29] showed that these methods significantly reduce analysis time, optimizing scientific research when compared to manual methods such as histomorphometric analysis. In the present study, ArcGIS promoted better delimitation of the areas of interest because each image is monitored manually, a fact that reduced the errors observed for the more automated TensorFlow method. Furthermore, comparison of TensorFlow and ArcGIS showed that, although the former did not include all areas of interest, the results were similar to those obtained with ArcGIS, which included all areas. Nevertheless, TensorFlow could still be used to measure areas of differentiation considering the similarities between the results.

TensorFlow is designed for machine learning-based data automation of biomedical processes, while ArcGIS is used in civil engineering for mapping and spatial analysis, and is therefore considered a method never used to measure areas of cellular differentiation. These methods showed equivalent performance in measuring osteogenic differentiation from images. Both methods can be used to optimize image analysis, ensuring speed and automation in scientific research, unlike manual methods such as histomorphometric analysis which, although accepted in the literature, are generally slower, as demonstrated by Miranda et al. [5].

In addition to providing speed and automation, ArcGIS and TensorFlow demonstrate high accuracy, as the precision of the models is evaluated using RMSE and the IoU metric, respectively. In terms of RMSE, smaller values are better, and the $R^2$ should be greater than 0.75; in this study, the $R^2$ was 0.95 and the RMSE was 0.05 or less, which are considered very satisfactory. A high and very precise IoU value (close to 1) indicates that the model's segmentation is close to the real data, in accordance with the gold standard method, which in this case was 0.94, confirming the high accuracy of the results.

Finally, analysis of osteogenic differentiation showed equivalent areas in G5 (HANp) and G6 (HANp + PBM) for a period of 21 days. Considering the lack of studies combining HANp and PBM, this is the first study reporting satisfactory results. The findings suggest the need for more detailed studies on the benefits of this combination, including gene expression analysis of hUC-MSCs treated with HANp and PBM to further elucidate its effects.

## Conclusion

Therefore, this study aimed to evaluate, in vitro, the effect of PBM combined or not with 30 nm hydroxyapatite nanoparticles (HANp) on the osteogenic differentiation of dhUC-MSCs through morphometric analysis using artificial intelligence programs (TensorFlow and Arc-GIS). Considering the parameters used in this study, it is suggested that HANp, whether combined with PBM or not, may be a promising alternative to enhance the cellular viability and osteogenic differentiation of hUC-MSCs. These results indicate that these therapies are important for obtaining an adequate number of cells that can be used for bone regeneration in in vivo studies and subsequent clinical trials, contributing to future clinical perspectives in the field of tissue engineering.

## Acknowledgments

We would like to thank the Prof. Oleg Vladimirovich Krasilnikov Laboratory of Biophysica and Stem Cells, Department of Biophysics and Radiobiology, Federal University of Pernambuco (UFPE). The PBM device for cell irradiation was kindly provided by the Biophotonics Laboratory (LABFOTONI/UPE) and Implant Laboratory of the University of Pernambuco (UPE).

## Author Contributions

**Conceptualization:** Marleny Elizabeth Márquez de Martínez Gerbi.

**Formal analysis:** Edson Luiz Pontes Perger, Mávio Eduardo Azevedo Bispo.

**Funding acquisition:** Jéssica Meirinhos Miranda, Marleny Elizabeth Márquez de Martínez Gerbi.

**Investigation:** Eloiza Leonardo de Melo, Jéssica Meirinhos Miranda, Vanessa Bastos de Souza Rolim Lima, Wyndly Daniel Cardoso Gaião, Braulio de Vilhena Amorim Tostes.

**Methodology:** Eloiza Leonardo de Melo, Jéssica Meirinhos Miranda, Vanessa Bastos de Souza Rolim Lima, Braulio de Vilhena Amorim Tostes.

**Project administration:** Claudio Gabriel Rodrigues, Márcia Bezerra da Silva, Severino Alves Júnior, Marleny Elizabeth Márquez de Martínez Gerbi.

**Resources:** Claudio Gabriel Rodrigues, Márcia Bezerra da Silva, Severino Alves Júnior, Edson Luiz Pontes Perger, Mávio Eduardo Azevedo Bispo, Marleny Elizabeth Márquez de Martínez Gerbi.

**Software:** Edson Luiz Pontes Perger, Mávio Eduardo Azevedo Bispo.

**Supervision:** Wyndly Daniel Cardoso Gaião, Claudio Gabriel Rodrigues, Márcia Bezerra da Silva, Severino Alves Júnior.

**Writing – original draft:** Eloiza Leonardo de Melo, Jéssica Meirinhos Miranda.

**Writing – review & editing:** Eloiza Leonardo de Melo, Jéssica Meirinhos Miranda.

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
