## [Decision Letter · Decision Letter 0]

18 Sep 2024

PONE-D-24-33995Effect of laser photobiomodulation combined with hydroxyapatite nanoparticles on the osteogenic differentiation of mesenchymal stem cells using artificial intelligence: An in vitro studyPLOS ONE

Dear Dr. Miranda,

Thank you for submitting your manuscript to PLOS ONE. After careful consideration, we feel that it has merit but does not fully meet PLOS ONE’s publication criteria as it currently stands. Therefore, we invite you to submit a revised version of the manuscript that addresses the points raised during the review process.

The reviewers have made extensive suggestions to improve the manuscript. Please consider and address all of the points in detail

We look forward to receiving your revised manuscript.

Kind regards,

Michael R Hamblin

Academic Editor

PLOS ONE

Journal requirements: 1. When submitting your revision, we need you to address these additional requirements. Please ensure that your manuscript meets PLOS ONE's style requirements, including those for file naming. The PLOS ONE style templates can be found at https://journals.plos.org/plosone/s/file?id=wjVg/PLOSOne_formatting_sample_main_body.pdf and https://journals.plos.org/plosone/s/file?id=ba62/PLOSOne_formatting_sample_title_authors_affiliations.pdf. 2. Please note that PLOS ONE has specific guidelines on code sharing for submissions in which author-generated code underpins the findings in the manuscript. In these cases, we expect all author-generated code to be made available without restrictions upon publication of the work. Please review our guidelines at https://journals.plos.org/plosone/s/materials-and-software-sharing#loc-sharing-code and ensure that your code is shared in a way that follows best practice and facilitates reproducibility and reuse.

Reviewers' comments:

Reviewer's Responses to Questions

**Comments to the Author**

1. Is the manuscript technically sound, and do the data support the conclusions?

Reviewer #1: Yes

Reviewer #2: No

2. Has the statistical analysis been performed appropriately and rigorously? 

Reviewer #1: Yes

Reviewer #2: Yes

3. Have the authors made all data underlying the findings in their manuscript fully available?

Reviewer #1: Yes

Reviewer #2: Yes

4. Is the manuscript presented in an intelligible fashion and written in standard English?

Reviewer #1: Yes

Reviewer #2: Yes

5. Review Comments to the Author

Reviewer #1: The manuscript entitled “Effect of laser photobiomodulation combined with hydroxyapatite nanoparticles on the osteogenic differentiation of mesenchymal stem cells using artificial intelligence: An in vitro study”.The aim to evaluate in vitro the effect of laser photobiomodulation (PBM; 660 nm, 10 mW, 28 2.5 J/cm², spot size of 0.08 cm²), combined or not with 30-nm hydroxyapatite nanoparticles (HANp), on the osteogenic differentiation of human umbilical cord mesenchymal stem cells (hUC-MSCs) by morphometric analysis using artificial intelligence programs (TensorFlow and ArcGIS).

Below are some suggestions:

In the Abstract:

- I suggest the authors reorganize the aim, the photobiomodulation protocol should be included in the methodology, it would be more appropriate;

- insert some morphometric data (values) in the results.

In the Introduction:

- The introduction is clear and objective.

In the Materials and Methods:

1.Irradiation of hUC-MSCs:

- The authors could explain the choice of low-level InGaAIP laser?

- It is necessary to insert a table fully describing the entire photobiomodulation protocol used...for example: power density, energy density, irradiance.... is extremely important.

2.Experimental Groups:

- I suggest the authors include an experimental design: experimental groups, biomaterials used, photobiomodulation....

- There also needs to be a paragraph describing the groups

In the Results:

- the legend of figure 1 needs to be better described

- Authors can insert the quality of the arrows in the images for better visibility.

In the Discussion:

- I suggest starting the discussion, in the first paragraph, with a contextualization of the manuscript, objective, as well as the main results.

In the Conclusion:

- I suggest inserting a conclusion with final considerations, including the objective of the research, its main results according to the objective and future clinical perspectives.

Reviewer #2: The main focus and novelty of this study is on the osteogenic differentiation of hUC-MSCs cells after stimulation with laser radiation and hydroxyapatite nanoparticles, which has been done using machine learning and cell shape analysis. Several controversial questions arise.

- TensorFlow and ArcGIS are used with what precision and accuracy do they differentiate the photos?

- It seems that only examining the shape of the cell cannot confirm its definitive differentiation. Are there similar papers that have only used this method to investigate cell differentiation? References 23 to 27 mentioned in the discussion did not use TensorFlow and ArcGIS to investigate osteoblast cell differentiation. Therefore, it seems that the use of machine learning instead of molecular methods cannot be a definitive method for investigating cell differentiation.

- The results of the figure related to Mineralization production (Alizarin Red staining) must be reported quantitatively. What is the reason that mineralization has decreased in some groups on day 21 compared to day 14?

- In the text of the article, it is said that the group of laser radiation together with nanoparticle has the same effect as nanoparticle alone. So can we say that laser radiation has no effect? But in Figure 1, it seems that the laser group alone has the same effect as the nanoparticle group alone. It seems that the results are not interpreted correctly.

- The caption of Figure 1 is incomplete. If FBM is used in Figure 1, but nowhere in the caption of the figure or in the text of the article is it explained what FBM is. The figure grouping should be the same as the grouping in the text. In this form, all treatment groups have almost the same results and have increased compared to the control group. Did the laser radiation group alone have the same effect as the HANp group alone?

- The results of TensorFlow program for all groups according to time after osteogenic induction seem contradictory and are not properly explained. If the level of differentiation in the groups that received laser or nanoparticles or the combination of two stimuli is similar to the group that received BMP, then why is this result not seen in the graph on day 21? Figures 4 and 5 should be explained and justified in detail. It seems that the graphs are not understood correctly.

- In order to conclude that HANp + laser can replace BMPs, it is necessary to carry out detailed genetic analysis.

- The quality of the Figures are very low

6. PLOS authors have the option to publish the peer review history of their article (what does this mean?). If published, this will include your full peer review and any attached files.

Reviewer #1: No

Reviewer #2: No

---

## [Author Response · Author response to Decision Letter 0]

17 Oct 2024

October 17, 2024

Michael R Hamblin

Academic editor

PLOS ONE

Dear Dr. Hambli,

We are submitting a revised version of the manuscript entitled “Effect of Laser Photobiomodulation Combined with Hydroxyapatite Nanoparticles on the Osteogenic Differentiation of Mesenchymal Stem Cells Using Artificial Intelligence: An In Vitro Study” (PONE-D-24-33995), along with a letter containing our point-by-point responses to the reviewers’ comments. We sincerely thank the editors and reviewers for their careful evaluation of our manuscript. We have thoughtfully considered their suggestions and addressed all comments. The changes in the text are highlighted in red font.

We would like to confirm that all authors have approved the submission of this manuscript version. The material submitted to PLOS ONE is not under consideration for publication elsewhere.

Respectfully,

Jéssica Meirinhos Miranda, DDS, PhD Student

Dear Editor,

First of all, I would like to thank you for the opportunity you gave me to revise the manuscript “Effect of laser photobiomodulation combined with hydroxyapatite nanoparticles on the osteogenic differentiation of mesenchymal stem cells using artificial intelligence: An in vitro study” (PONE-D-24-33995).

Reviewer #1: The manuscript entitled “Effect of laser photobiomodulation combined with hydroxyapatite nanoparticles on the osteogenic differentiation of mesenchymal stem cells using artificial intelligence: An in vitro study”.The aim to evaluate in vitro the effect of laser photobiomodulation (PBM; 660 nm, 10 mW, 28 2.5 J/cm², spot size of 0.08 cm²), combined or not with 30-nm hydroxyapatite nanoparticles (HANp), on the osteogenic differentiation of human umbilical cord mesenchymal stem cells (hUC-MSCs) by morphometric analysis using artificial intelligence programs (TensorFlow and ArcGIS).

Below are some suggestions:

In the Abstract:

- I suggest the authors reorganize the aim, the photobiomodulation protocol should be included in the methodology, it would be more appropriate;

Response: The corrections were made as requested in the "Abstract" section, line 33, page 2.

- insert some morphometric data (values) in the results.

Response: The corrections were made as requested in the "Abstract" section, line 41, page 2.

In the Introduction:

- The introduction is clear and objective.

In the Materials and Methods:

1.Irradiation of hUC-MSCs:

- The authors could explain the choice of low-level InGaAIP laser?

Response: This laser was chosen because it is already well-established in the literature that it accelerates the regenerative process of tissues due to its biomodulatory effect, as reported in the "Materials and Methods" section, subsection “Irradiation of hUC-MSCs”, pages 6-7, lines 147-150, and in the "Discussion" section, paragraph 3, lines 370-380, page 17. Furthermore, in our biophotonics research group, positive results have been observed in previous comparative studies after using this laser on mesenchymal stem cells, as verified in the studies by Miranda et al., 2020; Soares et al., 2015; and Zaccara et al., 2015, for example.

Miranda JM, de Arruda JAA, Moreno LMM, Gaião WDC, do Nascimento SVB, Silva EVS, et al. Photobiomodulation Therapy in the Proliferation and Differentiation of Human Umbilical Cord Mesenchymal Stem Cells: An In Vitro Study. J Lasers Med Sci. 2020;11:469-474. 

SOARES, Diego Moura et al. Effects of laser therapy on the proliferation of human periodontal ligament stem cells. Lasers in medical science, v. 30, p. 1171-1174, 2015.

ZACCARA, Ivana Maria et al. Effect of low-level laser irradiation on proliferation and viability of human dental pulp stem cells. Lasers in medical science, v. 30, p. 2259-2264, 2015.

- It is necessary to insert a table fully describing the entire photobiomodulation protocol used...for example: power density, energy density, irradiance.... is extremely important.

Response: The table related to the photobiomodulation protocol has been included as requested in the "Materials and Methods" section, subsection "Irradiation of hUC-MSCs," page 7.

2.Experimental Groups:

- I suggest the authors include an experimental design: experimental groups, biomaterials used, photobiomodulation....

Response: A topic “Experimental design” was included, in which an explanatory figure was created about this subject as requested in the topic “Materials and Methods”, subtopic “Experimental design”, page 8.

- There also needs to be a paragraph describing the groups

Response: The paragraph describing the groups was added to the paper as requested in the "Materials and Methods" section, "Experimental groups" subsection, page 7-8.

In the Results:

- the legend of figure 1 needs to be better described

Response: The corrections were made as requested in the "Results" section, subsection "MTT assay," page 12. This figure became figure 2, as a figure (Fig 1. Experimental design) was added before this one.

- Authors can insert the quality of the arrows in the images for better visibility.

Response: The corrections have been made to all the images as requested.

In the Discussion:

- I suggest starting the discussion, in the first paragraph, with a contextualization of the manuscript, objective, as well as the main results.

Response: The corrections were made as requested in the "Discussion" section, first paragraph, line 363-371, page 16-17.

In the Conclusion:

- I suggest inserting a conclusion with final considerations, including the objective of the research, its main results according to the objective and future clinical perspectives.

Response: The corrections were made as requested in the "Conclusion" section, lines 473-481, page 21.

Reviewer #2: The main focus and novelty of this study is on the osteogenic differentiation of hUC-MSCs cells after stimulation with laser radiation and hydroxyapatite nanoparticles, which has been done using machine learning and cell shape analysis. Several controversial questions arise.

- TensorFlow and ArcGIS are used with what precision and accuracy do they differentiate the photos?

Response: These are programs that exhibit accuracy and precision, as in ArcGIS the RMSE (Root Mean Square Error) has the same unit as the predicted values. Therefore, it's essential to understand the significance of RMSE when comparing it with the predicted values. To determine if it is good or not, the Dispersion Index (SI) is calculated, which is simply the RMSE divided by the average of the observed values: SI = (RMSE/average observed value) * 100%. This makes it easier to assess whether it is a good model. If SI < 10%, it is a good model; if SI < 5%, it is a very good model. According to the Ashrae standard, a predictive model should have an R² (coefficient of determination) greater than 0.75 and an SI lower than 30% when considering annual data, and 10% for hourly or monthly data. In terms of RMSE, the smaller, the better. It’s similar to defining an R² for calibration at 0.999, where RMSE values below 0.1 are very satisfactory. The adjusted R-squared depends on the criteria defined for the model or test; generally, a value above 0.6 is good, while values of 0.8 and above represent a very good model, especially when close to 1. The model presented RMSE values of 0.05 and an R² of 0.95. TensorFlow, along with convolutional neural networks, can generate models that learn to identify hierarchies of features, ranging from simple edges to complex representations of biological structures. Each analyzed image is transformed into a set of feature maps, highlighting the regions that the neural network deems most relevant based on the patterns learned from the provided images. The neural network then reduces the dimensionality of the feature maps, selecting only the most prominent patterns. In this way, a pixel-by-pixel segmentation mask is generated, where each pixel of the image is labeled as belonging or not belonging to the class of interest (e.g., osteogenic differentiation region). To define the model's precision, we measure the overlap between the model's segmentation and the ground truth (gold standard segmentation result) using IoU (Intersection over Union), calculating the ratio of the intersection to the union of the segmented pixels. A high and very precise IoU value (close to 1) indicates that the model's segmentation closely aligns with the real one, according to the gold standard method, which in this case was 0.94, indicating high precision of the results. A summarized explanation of the precision of the models is found in the "Materials and Methods" section, subsection "Machine Learning using the TensorFlow and ArcGIS Programs," first, third, and fifth paragraphs, pages 10 and 11. Precision was also discussed in the "Discussion" section, paragraph 13, page 20-21.

- It seems that only examining the shape of the cell cannot confirm its definitive differentiation. Are there similar papers that have only used this method to investigate cell differentiation? References 23 to 27 mentioned in the discussion did not use TensorFlow and ArcGIS to investigate osteoblast cell differentiation. Therefore, it seems that the use of machine learning instead of molecular methods cannot be a definitive method for investigating cell differentiation.

Response: Cell differentiation occurs through the use of differentiation inducers such as BMPs, which are responsible for osteogenic differentiation, a concept well-established in the literature. The TensorFlow and ArcGIS programs were used for the identification, measurement, and quantification of mineralized tissue recognized by AI (machine learning), given that preliminary tests were conducted by feeding data into the software for the identification of already confirmed mineralized tissues. The corrections were made as requested in the "Materials and Methods" section, subsection "Machine Learning using the TensorFlow and ArcGIS Programs," paragraph 1, page 10. Regarding references from other authors, this work is original, as the use of TensorFlow and ArcGIS software has never been employed in the literature for measuring areas of cell differentiation. However, these programs are well-established in biomedical processes and civil engineering, respectively, giving this study a novel character, as explained in the "Discussion" section, paragraph 12, page 20. Other studies using PCR have shown similar results to those presented by artificial intelligence, which drastically reduces high costs associated with molecular methods, for example. Thus, AI can visualize/quantify mineralized areas in a manner similar to studies using PCR, yielding results comparable to studies that did not employ this analysis.

- The results of the figure related to Mineralization production (Alizarin Red staining) must be reported quantitatively. What is the reason that mineralization has decreased in some groups on day 21 compared to day 14?

Response: Tables 3 and 4 related to the quantitative data (mean and standard deviation) have been added to the "Results" section, subsection "Morphometric Analysis using TensorFlow and ArcGIS," pages 15 and 16. The decrease in mineralization from 14 days to 21 days in some groups can be attributed to the progressive reduction in the capacity for self-renewal and the consequent specialization of the cell into a functional cell, in this case, towards osteoblasts for the groups that did not receive the HANp. This information has been included in the "Discussion" section, paragraph 10, page 19.

- In the text of the article, it is said that the group of laser radiation together with nanoparticle has the same effect as nanoparticle alone. So can we say that laser radiation has no effect? But in Figure 1, it seems that the laser group alone has the same effect as the nanoparticle group alone. It seems that the results are not interpreted correctly.

Response: The group that used only nanoparticles (G-5: HANp) showed greater efficacy compared to the group that combined the nanoparticle with laser treatment (G-6: HANp + PBM), although there was no statistically significant difference at the 48 and 72-hour time points. When comparing the laser group (G-4: PBM) and nanoparticles in isolation (G-5: HANp), both were effective in accelerating cell proliferation when compared to the control group, indicating that laser radiation was also effective, just like the nanoparticles. The interpretation of Figure 1 has been modified and is included in the "Results" section, subsection "MTT Assay," page 12.

- The caption of Figure 1 is incomplete. If FBM is used in Figure 1, but nowhere in the caption of the figure or in the text of the article is it explained what FBM is. The figure grouping should be the same as the grouping in the text. In this form, all treatment groups have almost the same results and have increased compared to the control group. Did the laser radiation group alone have the same effect as the HANp group alone?

Response: The caption of Figure has been adjusted as requested, presenting the same grouping as described in the text, lines 274-276, page 12. When comparing the laser group (G-4: PBM) and nanoparticles in isolation (G-5: HANp), both were effective in accelerating cell proliferation compared to the control group, indicating that laser radiation also increased cell proliferation, just like the nanoparticles. However, the group that combined the nanoparticles showed superior proliferative effects compared to the laser. The interpretation of Figure 1 has been modified and is included in the "Results" section, subsection "MTT Assay," page 12.

- The results of TensorFlow program for all groups according to time after osteogenic induction seem contradictory and are not properly explained. If the level of differentiation in the groups that received laser or nanoparticles or the combination of two stimuli is similar to the group that received BMP, then why is this result not seen in the graph on day 21? Figures 4 and 5 should be explained and justified in detail. It seems that the graphs are not understood correctly.

Response: The decrease in mineralization at 21 days for the groups with BMPs can be attributed to the progressive reduction in self-renewal capacity and the consequent specialization of the cell into a functional cell, in this case towards osteoblasts, suggesting that there may have been cellular apoptosis. However, the groups that used the nanoparticle either in isolation or combined with PBM showed an increase in cell differentiation, indicating that the nanoparticle progressively and rapidly stimulated cell differentiation, whether associated with PBM therapy or not, as evidenced by the presence of mineralized tissue after 21 days. Additionally, the HANp groups demonstrated a positive effect on the adhesion and differentiation of stem cells and exhibited excellent biological properties compared to the other groups analyzed. This information has been added to the "Discussion" section, paragraph 10, page 19.

- In order to conclude that HANp + laser can replace BMPs, it is necessary to carry out detailed genetic analysis.

Response: This article concludes that the results suggest the need for more detailed studies on the benefits of combining HANp and laser photobiomodulation (PBM), including the analysis of gene expression in hUC-MSCs to better elucidate their effects, as mentioned in the last paragraph of the "Discussion" section, page 21. However, HANp, whether associated with PBM or not, increased osteogenic proliferation and differentiation, outperforming the groups that used BMPs. These are already well-established in the literature as gold standard growth factors for tissue regeneration, but they come with high costs that limit their application and justify the search for alternative experimental therapies, as explained in the last paragraph of the "Discussion," paragraph 6, page 18.

- The quality of the Figures are very low

Response: The resolution quality of the figures has been improved as requested; we are therefore resending the new files corresponding to the figures.

---

## [Decision Letter · Decision Letter 1]

31 Oct 2024

Effect of laser photobiomodulation combined with hydroxyapatite nanoparticles on the osteogenic differentiation of mesenchymal stem cells using artificial intelligence: An in vitro study

PONE-D-24-33995R1

Dear Dr. Miranda,

We’re pleased to inform you that your manuscript has been judged scientifically suitable for publication and will be formally accepted for publication once it meets all outstanding technical requirements.

Kind regards,

Michael R Hamblin

Academic Editor

PLOS ONE

Additional Editor Comments (optional):

Reviewers' comments:

Reviewer's Responses to Questions

**Comments to the Author**

1. If the authors have adequately addressed your comments raised in a previous round of review and you feel that this manuscript is now acceptable for publication, you may indicate that here to bypass the “Comments to the Author” section, enter your conflict of interest statement in the “Confidential to Editor” section, and submit your "Accept" recommendation.

Reviewer #1: All comments have been addressed

2. Is the manuscript technically sound, and do the data support the conclusions?

Reviewer #1: Yes

3. Has the statistical analysis been performed appropriately and rigorously? 

Reviewer #1: Yes

4. Have the authors made all data underlying the findings in their manuscript fully available?

Reviewer #1: (No Response)

5. Is the manuscript presented in an intelligible fashion and written in standard English?

Reviewer #1: Yes

6. Review Comments to the Author

Reviewer #1: I would like to thank the authors of the manuscript who have made all the suggestions requested in the review, so I indicate accept inpresent form.

7. PLOS authors have the option to publish the peer review history of their article (what does this mean?). If published, this will include your full peer review and any attached files.

Reviewer #1: No

---

## [Editor Report · Acceptance letter]

5 Nov 2024

PONE-D-24-33995R1 

PLOS ONE

Dear Dr. Miranda, 

I'm pleased to inform you that your manuscript has been deemed suitable for publication in PLOS ONE. Congratulations! Your manuscript is now being handed over to our production team.

Kind regards, 

on behalf of

Dr. Michael R Hamblin 

Academic Editor

PLOS ONE